# Expression and Polymorphism of TSLP/TSLP Receptors as Potential Diagnostic Markers of Colorectal Cancer Progression

**DOI:** 10.3390/genes12091386

**Published:** 2021-09-06

**Authors:** Abdelhabib Semlali, Mikhlid H. Almutairi, Abdullah Alamri, Narasimha Reddy Parine, Maha Arafah, Majid A. Almadi, Abdulrahman M. Aljebreen, Othman Alharbi, Nahla Ali Azzam, Riyadh Almutairi, Mohammad Alanazi, Mahmoud Rouabhia

**Affiliations:** 1Groupe de Recherche en Écologie Buccale, Faculté de Médecine Dentaire, Université Laval, 2420 Rue de la Terrasse, Local 1758, Québec, QC G1V 0A6, Canada; mahmoud.rouabhia@fmd.ulaval.ca; 2Zoology Department, College of Science, King Saud University, Riyadh 11451, Saudi Arabia; malmutari@ksu.edu.sa; 3Department of Biochemistry, College of Science, King Saud University, Riyadh 11451, Saudi Arabia; abdullah@ksu.edu.sa (A.A.); nparine@ksu.edu.sa (N.R.P.); ralmutairi@ksu.edu.sa (R.A.); msanazi@ksu.edu.sa (M.A.); 4Pathology Department, College of Medicine, King Saud University, Riyadh 11451, Saudi Arabia; marafah@ksu.edu.sa; 5Division of Gastroenterology, College of Medicine, King Saud University, Riyadh 11451, Saudi Arabia; majid.almadi@gmail.com (M.A.A.); amaljebreen@gmail.com (A.M.A.); alharbiothman@hotmail.com (O.A.); nazzam@ksu.edu.sa (N.A.A.)

**Keywords:** TSLP, TSLPR, polymorphisms, colorectal cancer

## Abstract

Colorectal cancer (CRC) is the third most common malignancy and the fourth leading cause of cancer-related mortality worldwide. Inflammation is considered as a critical driver for CRC development and growth. We investigated the association between polymorphisms/expression levels of thymic stromal lymphopoietin (TSLP) /TSLP receptors and CRC risk in Saudi population. DNA samples were isolated from blood samples from 220 participants. Case subjects were 112 patients diagnosed with CRC, while control subjects were 108 healthy individuals, who were not diagnosed with any type of malignancy. We selected two single nucleotide polymorphisms (SNPs) located in the thymic stromal lymphopoietin gene (*rs10043985* and *rs2289276*), three SNPs in TSLP receptor gene (*TSLPR*; *rs36139698*, *rs36177645*, and *rs36133495*), and two other SNPs in interleukin-7 receptor gene (*IL-7R*; *rs12516866* and *rs1053496*), and designated these SNPs for a case-control genotyping study. The gene expression was analyzed using quantitative RT-PCR and immunohistochemistry assays array on 20 matching colorectal cancer/normal tissues. mRNA expressions and protein levels of TSLP, TSLPR-α subunit, and IL-7R-α subunit showed a 4-fold increase in colon cancer tissues when compared to normal colon tissues. Furthermore, two SNPs (rs10043985 of TSLP and rs1053496 of IL-7R) showed statistically significant correlations with CRC susceptibility. Interestingly, only rs10043985 showed a statistically significant association (*p* < 0.0001) in the genotypic and phenotypic levels with CRC for all clinical parameters (age, gender, and tumor location) tested. However, IL-7R rs1053496 genotyping results presented a significant correlation (*p* < 0.05) in male CRC patients and in individuals under 57 years of age. TSLP rs2289276, IL-7R rs12516866, and all TSLPR variants did not display any significant genotypic or phenotypic correlations in all tested clinical parameters. This study identified that TSLP rs10043985 and IL-7R rs1053496 SNPs, and the expression levels of TSLP and TSLPR-α subunit, can be used as markers for CRC development and treatment. However, additional investigations are required on larger group of patients from diverse ethnicities to confirm the genetic association of these variants to CRC.

## 1. Introduction

Colorectal cancer (CRC) is the third most common type of malignancy and the fourth leading cause of cancer-related mortality among global population [1,2]. According to the American Cancer Society statistics, around 130,000 new cases of CRC are identified every year and almost 50,000 people die due to CRC [3]. In the Kingdom of Saudi Arabia (KSA), the frequency of occurrence of CRC is much lower but it seems to be increasing every year due to changes in the lifestyle [4]. In the KSA population, CRC mortality ranks first among men (8.8%) and second among women (7.6%), which represents 11.3% of death caused by all cancer types [5]. CRC is an age dependent disease, with 90% of cases occurring in individuals above 50 years of age, and the incidence is 50-fold higher in populations who are 60–78 years of age than people younger than 40 years [6]. Obesity [7], excessive alcohol consumption [8], excessive red meat and processed meat consumption [9], a high fat diet [10], smoking, low socioeconomic status [11], and physical inactivity are factors that might lead to the disease [12]. CRC is a complex inflammatory disease and a multifactorial disorder. However, the risk factors of CRC are determined by the interactions between inherited genes and environmental exposures [13,14,15]. The role of inflammation in cancer initiation has been linked with cancer cell proliferation and migration/invasion among tumor cells of different types [15]. Also, CRC is more frequent in people suffering from inflammatory bowel disease (IBD) [16]. Secreted cytokine proteins provide signals between immune cells to coordinate the inflammatory response. Certain cytokines, such as Interleukin-1 (IL-1), Interleukin-6 (IL-6), Tumor necrosis factor (TNF), and Thymic stromal lymphopoietin (TSLP), act to broadly induce an inflammatory response while others act on specific types of immune cells [17].

TSLP is a novel IL-7–like cytokine, originally cloned from a murine thymic stromal cell line [18]. It is one of the members of the IL-2 cytokine family [19]. The human TSLP gene is located on chromosome 5q22.1 next to the atopic cytokine cluster of chromosome 5q31 [20]. Like IL-7, TSLP is identified as a four-helix bundle cytokine. Even though there is poor amino acid sequence identity between murine and human TSLPs (only 43% amino acid sequence identity), they share significant similarities at the functional level [19]. TSLP signaling requires a heterodimeric receptor complex made up of the TSLP receptor (TSLPR) subunit and interleukin-7 receptor (IL-7R) α subunit [21,22]. The complex functions to activate *STAT1*, *STAT3*, *STAT4*, and *STAT5* which promotes the proliferation, development, differentiation, migration, and death of apoptotic cells, depending on the type of stimuli and cells [23,24]. It is also known to play a pivotal role in atopic dermatitis (AD), asthma, cancer, and other inflammatory diseases. TSLP gene knock down or TSLPR deficiency decreased breast cancer cell growth and metastasis, indicating a critical role of TSLP in cancer metastasis [25]. TSLP is predominantly expressed in epithelial cells of the intestines, skin (keratinocytes), and lung (small airway epithelial cells) [19,22,26]. Gounni et al. have detected correlations between enhanced TSLP expression and both the expression of Th2-attracting chemokines and also with the disease severity in asthmatic airways [27]. Recent studies have demonstrated that immune responses play a fundamental role in tumor progression at different stages; for instance, initiation, invasion, and metastasis [26,28,29]. Other recent studies also indicated TSLP as a key molecule contributing to the cross talk between various cell types involved in lung, breast and pancreatic cancers [30,31]. However, the possible TSLP regulatory mechanisms underlying these cancers are not clear and varied in different tumors. Recently, several studies have shown the link between genetic variants in TSLP gene and susceptibility to diseases, such as, the 23 TSLP SNPs found in asthmatic individuals [32,33,34]. Another study discovered the role of a specific TSLP gene polymorphism in the reduction of immunoglobulin E (IgE) levels in response to cockroach allergy [35]. A number of studies have evaluated the various genetic polymorphisms in different populations. TSLP SNPs associated with asthma are rs1837253, rs17551370, and rs2289276 [35,36], which differ in the risk to cause the disease. These SNPs differ from one population to another [36]. In our previous study, we have reported that TSLP/TSLP receptors polymorphism were associated in breast cancer progression in Saudi population [37]. In this study, we aimed to identify more efficient biomarkers to facilitate early CRC diagnosis and treatment. Consequently, the goals of this study are to determine the distribution of TSLP SNPs (rs10043985 and rs2289276), TSLPR SNPs (rs36139698, rs36177645, and rs36133495) and two others SNPs of IL-7R (rs12516866 and rs1053496) in CRC patients in Saudi population, and also to compare the expressions of TSLP, TSLPR, and IL-7R at both mRNA and protein levels in CRC tissues when compared to normal colon tissues. The choose of this SNPs is due to our previous study and in literature review and to their localization in TSLP/TSLPR genes.

## 2. Materials and Methods

### 2.1. Study Population and Data Collection

In this cross-sectional study, samples were collected from 220 individuals, including 112 CRC cases and 108 normal colon tissues from Saudi Arabian population. Blood and tissue samples of all the individuals including their recorded clinical data were obtained from King Khalid University Hospital (KKUH), Riyadh, Saudi Arabia, and this study was approved by the ethics committee of the College of Applied Medical Sciences at King Saud University (KSU), Riyadh (project E-12-596, 12/3352/IRB). We confirm that all methods were performed in accordance with the relevant guidelines and regulations. However, informed written consent and a self-administered questionnaire regarding sociodemographic character (e.g., age, family history of cancer, etc.), lifestyle (e.g., smoking habits and alcohol intake), and personal medical history were collected from all the participants. The age group of the entire individual studied was between 45 to 88 years of age. All tissue samples were stored in RNAlater solution (Ambion^®^ Life Technologies, Foster City, CA, USA) to avoid RNA degradation and for further DNA and RNA extraction steps. Blood samples (4 mL) were collected from each individual and then stored in EDTA tubes at −80 °C. Tissue samples were collected for immunohistochemistry studies and the blood samples for genotyping assays.

### 2.2. Extraction of Total RNA from Colon Cancer and Normal Colon Tissues

For RNA extraction, tissues were homogenized using Medic Tools (Zürich, Switzerland) homogenizer. Intact total RNA isolation for 20 CRC tissues and 20 matched normal colon tissue samples were obtained by using the RNeasy Plus Mini Kit (Qiagen, Hilden, Germany) according to the manufacturer’s protocol. Briefly, 10 µL β-mercaptoethanol (β-Me) and 350 µL of buffer RLT were added to each homogenized tissue in 1.5 mL collection tubes with constant mixing required during each step, and then, 350 μL of 70% ethanol was added. After mixing, 700 µL was transferred to an RNeasy Plus Mini Kit column. The column was washed with 700 µL of washing buffer and 500 µL RPE (twice) and during every addition step, mixing and centrifugation were done. In total, 50 µL of elution buffer was added twice with incubation of 10 min during each step. Spectrophotometric analysis (NanoDrop, Thermo Scientific, London, UK) and agarose gels were done to assess RNA quality.

### 2.3. Complementary DNA (cDNA) Synthesis

Single stranded cDNA was synthesized from the purified RNA using random primers and the high capacity cDNA reverse transcription kit (Applied Biosystems, Warrington, PA, USA) according to the manufacturer’s protocol. One microgram of the total isolated RNA was mixed with 10 µL of the master mix solution in an Eppendorf tube to perform polymerase chain reaction (PCR). Briefly, the conditions applied for cDNA synthesis were as follows: 10 min at 25 °C, 2 h at 37 °C, 5 min at 85 °C, and 10 min at 4 °C. Finally, the samples were then stored at −20 °C for further use.

### 2.4. Quantitative RT-PCR (qRT-PCR)

Gene expression levels were assessed by qRT-PCR in triplicates using SYBR Green Supermix (Applied Biosystems, Hercules, CA, USA). The experimental conditions followed are as explained by Langner et al. [38] with the only difference being in the plate type used; 96-well fast reaction plates (Applied Biosystems, Darmstadt, Germany) were used in our experiments. Briefly, 25 µL reactions include 12.5 µL SYBR Green master mix, 0.5 µL primers (F+R), 7 µL distilled water, and 5 µL of diluted cDNA synthesized from the isolated RNA (2 µg) (dilution was 1/10), placed in 96-well fast reaction plates (Applied Biosystems, Darmstadt, Germany). The PCR results were analyzed on a 7500 Fast Real-Time PCR System (Applied Biosystems, Hercules, CA, USA). The conditions of qRT-PCR amplification are: for the hold step, 2 min at 50 °C, 5 min at 95 °C; for the PCR step, 30 s at 95 °C, 45 s at 72 °C and 15 min at 95 °C; and for the melt curve step, 30 s at 95 °C, 45 s at 60 °C and 15 min at 95 °C. The only condition that was different for each primer was the annealing temperature which was set at 60 °C for Glyceraldehyde 3-phosphate dehydrogenase (GAPDH) and IL-7R and at 62 °C for TSLP and TSLPR. GAPDH was used as a positive control for normalization of qRT-PCR results. The amounts of relative mRNA transcripts were measured with threshold cycle (CT) value using the accompanying Applied Biosystems software. The results were analyzed by Livak relative expression method. The forward and reverse primers used for each gene (*TSLP*, *TSLPR*, *IL-7R*, or *GAPDH*) are listed in Table 1.

### 2.5. Immunohistochemistry Array (IHC Array) and Histology Analysis

IHC array were performed by use CRC tissues and matching normal colon tissue specimens (70 CRC tissues and 70 matched normal colon tissues). The tissues were cut and placed perfectly to fit inside the cassette and are treated with 10% neutral buffered formalin. Paraformaldehyde solution was used to fix the tissues. After the fixation step, tissue processing was done by adding 10% formalin solution twice, followed by increasing concentrations of alcohol at 70%, 90%, 95%, and 100% xylene twice with each step lasting about 1 h at 40 °C and the specimens were embedded in paraffin wax. The embedded tissue specimens were sectioned at about 3-μm thickness with a microtome. The sections were then transferred into a 20% alcohol container and then pre-heated water at 47 °C was added and incubated for 25–30 min in a dry oven at 65 °C. Dewaxing by the verses steps of tissue processing is performed for 1 min for each step. Antigen retrieval was done by the pH and the heat of steamer after that treat it with hydrogen peroxide and protein blocking before the antibody added. The slides were incubated for 1 h at 37 °C with one of the antibodies: anti-human-TSLP, anti-human-TSLPR, or anti-human-IL7R (1:100 dilution, Santa Cruz Biotechnology, Dallas, TX, USA). This was followed by incubation for 15 min at 37 °C with a secondary antibody ultraView Universal HRP multimer (1:200 dilutions, Santa Cruz Biotechnology, Dallas, TX, USA). Instead of adding the primary antibodies, PBS was added in negative control. The immunolocalized TSLP and TSLPR proteins were visualized using a copper-enhanced DAB reaction. Slides were then stained using Hematoxylin and Eosin stain, and were then visualized under a microscope. The intensities of TSLP, TSLPR, and IL-7R staining was assessed in a blind manner by two independent investigators and graded using a 5-point system (0, no signal; 1, weak; 2, moderate; 3, strong; 4, very strong; and 5, extremely strong). 

### 2.6. Extraction of DNA from Blood Specimens

Genomic DNA was obtained from 200 µL of freshly stored blood samples by using QIAmp DNA Blood Mini Kit (Qiagen, Valencia, CA, USA) according to manufacturer’s protocol. Briefly, the blood samples were treated with 20 µL of protease and 200 µL AL buffer, incubated for 10 min at 56 °C, mixed, centrifuged and placed in a spin column. In total, 500 µL of both the washing buffers (AW1 and AW2) was added to the column, mixed, and centrifuged after each addition. The elution was done twice in 50 µL AE buffer. DNA yield and purity was measured using NanoDrop 8000 spectrophotometer (Thermo Scientific, London, UK).

### 2.7. TaqMan Genotyping Assay

TSLP SNPs (rs10043985 and rs2289276), TSLPR SNPs (rs36139698, rs36177645, rs36133495) and IL-7R SNPs (rs12516866, and rs1053496) were genotyped using a TaqMan genotyping assay. The details of all SNPs analyzed are mentioned in Table 2. Every step in the assay was done according to the manufacturer’s recommendations. Briefly, the assay was conducted in 96-well Fast Reaction Plates (Applied Biosystems, Darmstadt, Germany). In total, 560 µL TaqMan Master Mix, 27 µL SNPs assay, 275 µL distilled water and 2 µL DNA was added to each well. The prepared plates were operated on 7500 Fast Real-Time PCR System (Applied Biosystems, Hercules, CA, USA). The conditions applied for qRT-PCR include a 30 s pre-read step at 60 °C, 10-min hold step at 95 °C, followed by 45 cycles of PCR reaction, which include 15 s denaturation steps at 95 °C, then a 1 min annealing step at 60 °C and a final 30 s extension step at 60 °C.

### 2.8. Statistical Analyses

Following the protocol mentioned in Semlali et al. [39,40], the genotype and allele frequencies were computed and checked for deviations with Hardy-Weinberg equilibrium. Case-control and other genetic comparisons were performed using chi-square tests and allelic odds ratios (ORs) and 95% confidence intervals (CIs) were calculated using Fisher’s exact test (two-tailed *p* values). For analyzing TSLP, TSLPR, and IL-7R expressions, the differences between two groups were assessed by an unpaired student *t*-test. All statistical analyses were performed using SPSS version 22.0 statistical software (Statistical Package for the Social Sciences, Chicago, IL, USA) for Windows. All *p* values less than 0.05 were considered statistically significant.

## 3. Results

### 3.1. Study Population Characterization

As shown in Table 3, clinical data parameters of the study subjects, either with CRC or healthy control individuals, were recorded. The clinical parameters include the total number of study population in each category, gender type, age of participants at diagnosis, and tumor localization of the CRC patients. A total number of 200 subjects were investigated composing of 112 CRC individuals and 108 healthy individuals. Our investigations on the general demographic characteristics of the donors determined that the gender ratio was 64 males (57.14%) and 48 females (42.86%) among the subjects with CRC. In the case of healthy subjects, 61 participants (56.48%) were males, and 47 participants (43.52%) were females. Furthermore, 61 (54.46%) CRC patients had malignancies localized in the colon, while 51 patients (45.54%) had CRC tumors localized in the rectum.

### 3.2. mRNA Expression and Protein Levels of TSLP/TSLPRs in Colon Cancer Tissue

We investigated 20 samples from colon cancer tissues and 20 samples from normal colon tissues from the same patient for TSLP expression. This comparison was performed by using qRT-PCR assay. The mRNA expressions of TSLP showed an increase of about 4-folds in colon cancer tissues compared to normal colon tissues (*p* < 0.001, Figure 1A). To confirm our result of the mRNA expression, we next investigated the protein levels of TSLP via IHC analysis. We used four biopsies from colon cancer patients and applied a specific antibody, anti TSLP (Santa Cruz Biotechnology, Dallas, TX, USA; dilution 1/100). By analyzing the staining intensity, it was observed that protein levels of TSLP were significantly higher in colon cancer tissues than in normal colon tissues (Figure 2A). mRNA expression and protein levels of TSLPR-α subunit in colon cancer tissue were about 4-fold higher in all colon cancer samples when compared to normal colon samples (*p* < 0.0001, Figure 1B). In the confirmation step by IHC, we used a specific antibody anti TSLPR (Santa Cruz Biotechnology, Dallas, TX, USA; dilution 1/100). Staining intensity of TSLPR-α subunit showed that protein levels of TSLPR-α subunit were significantly higher in colon cancer tissues than those in normal matching colon tissues (Figure 2B). In order, mRNA expression and protein levels of IL-7R-α subunit in colon cancer tissue were increased at a *p* < 0.05, as shown in Figure 1C. To confirm the mRNA expression result by using the by the IHC, we found that the IL-7R-α subunit protein was found at reduced levels in colon cancer tissues compared to normal colon tissues (Figure 2C).

### 3.3. Global Study of the Association between TSLP, TSLPR, and IL-7R Polymorphisms among CRC Patients

In the present study, three different genes with different SNPs were selected; two variants in TSLP (rs10043985 and rs2289276), three variants in TSLPR (rs36139698, rs36177645, and rs36133495), and two other variants in IL-7R (rs12516866 and rs1053496). These SNPs were examined to evaluate the risk of genetic variation in these three genes among 112 CRC subjects and 108 normal controls. Determination of the odds ratio (OR) in the analysis of the genotyping results was done using ancestral (homozygous) alleles in these genetic variations (SNPs) as references. A general comparison between the CRC and control, the allele distributions of the examined SNPs, as well as, the association analysis is shown in Table 4.

Within these SNPs, two SNPs showed a statistically significant association with CRC, *rs10043985* of *TSLP* gene and *rs1053496* of *IL-7R* gene. The genotype distribution of *rs10043985* SNP in *TSLP* are as follows: 93% AA and 7% AC in control, while 46% AA and 54% AC in cases. The heterozygous allele AC represents higher correlation, about 16-fold, with cases compared to the homozygous allele ‘AA’ (OR: 16.52; CI: 7.042–38.783; *p* < 0.0001). The combination of AC+CC genotypes was about 16-fold higher in cases compared to the wild type genotype (OR: 16.52; CI: 7.042–38.783; *p* < 0.0001). In addition, we also found a significant phenotypic association of rs10043985 SNP in cases. The distribution of the phenotype was 97% A and 3% C in control and 73% A and 27% C in cases, giving approximately an 10-fold increased association of the C phenotype with cases in comparison to the phenotype A (OR: 10.83; CI: 4.820–24.363; *p* < 0.0001). On the other hand, the genotype frequency of *rs1053496* SNP in *IL-7R* gene was 18% CC, 44% CT, and 38% TT in control, while 30% CC, 35% CT, and 35% TT in cases. The homozygous allele TT shows a significant correlation with CRC effect protected risk about half fold (OR: 0.467; CI: 0.226–0.968; *p* = 0.03903). The CT+TT genotypes were decreased in cases compared to the wild type genotype (OR: 0.496; CI: 0.255–0.963; *p* = 0.03640). For this SNP, we noticed a significant phenotypic association with cases. The allelic distribution was 37% C and 63% T in controls and 47% C and 53% T in cases (OR: 0.64; CI: 0.434–0.955; *p* = 0.02823). In this study; however, we have not seen any significant association between CRC and the TSLP rs2289276, IL-7R rs12516866, and all TSLPR SNPs. During the comparison between the control and the cases, we observed that the allele frequencies of TSLP rs2289276, IL-7R rs12516866, and all TSLPR SNPs were similar. In TSLP, rs2289276 showed that 44% CC, 48% CT, and 8% TT for control and 49% CC, 40% CT, and 11% TT in cases. While in TSLPR, rs36133495 distributed as 14% CC, 49% CT, and 37% TT in control and 14% CC, 46% CT, and 40% TT in cases (Table 4).

### 3.4. TSLP, TSLPR, and IL-7R Polymorphisms Associations with the CRC Patient’s Ages

In different studies, age plays a role in progression of diseases [40,41]. Therefore, to affirm this study, we have considered age as a parameter. According to the Saudi Cancer Registry, the median age for CRC to occur is 57 years [42]; therefore, we have classified subjects based on age as group A (below 57 years old) and group B (above 57 years old). The genotype analyses of the SNPs for group A and B compared to the control are described in Table 5 and Table 6, respectively. Only *rs10043985* SNP of *TSLP* gene showed a statistically significant association in the genotypic and phenotypic levels with CRC for both group A and group B. The genotype distribution in TSLP rs10043985 SNP as the following; group (A) 94% AA and 6% AC in controls, while 46% AA and 54% AC in cases. Comparison of the heterozygous allele AC with the wild homozygous allele AA resulted in about 17-fold higher correlation with cases (OR: 17.11; CI: 5.443–53.781; *p* < 0.0001). The combination of AC+CC genotypes was about 17-fold higher in cases, compared to the wild type genotype (OR: 17.11; CI: 5.443–53.781; *p* < 0.0001). In addition, this study also found a significant phenotypic association with cases. The distribution of the phenotype was 97% A and 3% C in control and 73% A and 27% C in cases, showed greater association between C phenotype with cases and the wild type about 11-fold (OR: 11.196; CI: 3.79–33.069; *p* < 0.0001). However, the distribution of group (B) for the same SNP rs10043985 was; 93% AA and 7% AC in controls, while 46% AA and 54% AC in cases. The allele AC represents about 15-fold higher correlation with cases compared to the allele AA (OR: 15.77; CI: 4.365–56.973; *p* < 0.0001). The combinations of AC+CC genotypes were about 15-fold greater in cases compared to the wild type genotype (OR: 15.77; CI: 4.365–56.973; *p* < 0.0001). In addition, we also found significant phenotypic association with cases. The distribution of the phenotype was 97% A and 3% C in control and 73% A and 27% C in cases, giving about 10-fold increased association of the C phenotype with cases, compared to the phenotype A (OR: 10.366; CI: 3.045–35.286; *p* < 0.0001). Not like the previous general compression (for Table 4), IL-7R genotype results were not significant except rs1053496 SNP. It showed a significant association with decreased CRC risk for phenotypic level in group A, which is not observed in group B. The T phenotype compared to the C phenotype, was more correlated (0.580-fold) to patients below 57 years of age (OR: 0.580; CI: 0.339–0.991; *p* = 0.04556). However, TSLP rs2289276, IL-7R rs12516866, and all TSLPR SNPs did not show any significant genotypic or phenotypic association with cases in both groups of ages (Table 5 and Table 6).

### 3.5. TSLP, TSLPR, and IL-7R Polymorphism Associations among the CRC Patients’ Gender

The frequency of occurrence of any serious disease differs in the ethnic group, age, environmental factor or gender [24,39,40,43]. In our study, we compared the CRC patients with gender (male and female) to assess the association of genetic variation in *TSLP*, *TSLPR* and *IL-7R* genes with the gender. We performed genotype analysis for each gender separately for the male group (A) and for the female group (B), which is described in Table 7 and Table 8, respectively. We observed that the *rs10043985* SNP of *TSLP* gene showing a statistically significant association in the genotypic and phenotypic levels with CRC for both male and female groups. The genotype distribution in TSLP as follows: for group (A) 93% AA and 7% AC in controls, while 42% AA and 58% AC in cases. The heterozygous allele AC represents approximately 19-fold higher correlation with cases compared to the homozygous allele AA (OR: 19.038; CI: 6.129–59.139; *p* < 0.0001). Furthermore, the combinations of AC+CC genotypes were about 19-fold higher in cases compared to the wild type genotype (OR: 19.038; CI: 6.129–59.139; *p* < 0.0001). In addition, we also found in this investigation a significant phenotypic association with cases. The distribution of the phenotype was 97% A and 3% C in control and 71% A and 29% C in cases, giving about a 11-fold increased association of the C phenotype with cases compared to the phenotype A reference allele (OR: 11.659; CI: 4.000–33.983; *p* < 0.0001). In contrast, the distribution for group (B) of rs10043985 was; 93% AA and 7% AC in controls, while 52% AA and 48% AC in cases. The allele AC represents about 13-fold higher correlation with cases compared to the allele AA (OR: 13.187; CI: 3.593–48.395; *p* < 0.0001). The combinations of AC+CC genotypes were about 13-fold higher in cases compared to the wild type genotype (OR: 13.187; CI: 3.593–48.395; *p* < 0.0001). Additionally, we also found a significant phenotypic association with cases. The distribution of the phenotype was 97% A and 3% C in control and 76% A and 24% C in cases, resulted in increasing the association of the C phenotype with cases about 9-fold compared to the phenotype A (OR: 9.347; CI: 2.699–32.374; *p* < 0.0001).

For IL-7R rs1053496 SNP genotype, the heterozygous allele CT was not associated with the cases. The distribution was; 17% CC, 37% CT, and 46% TT in control, while 35% CC, 32% CT, and 33% TT in cases. The heterozygous allele ‘CT’ showed about 0.4-fold (OR: 0.409; CI: 0.152–1.104; *p* = 0.07464). However, the homozygous allele TT genotypes were 0.344-fold protected the risk of cases compared to the wild type genotype (OR: 0.344; CI: 0.130–0.905; *p* = 0.02824). The CT+TT genotypes were 0.373-fold decrease in cases compared to the wild type genotype (OR: 0.373; CI: 0.154–0.902; *p* = 0.02572). In addition, this SNP exhibited a significant phenotypic association decrease with cases for this SNP. The T phenotype, compared to the C phenotype were distributed as 35% C and 65% T in control and 51% C and 49% T in cases (OR: 0.526; CI: 0.310–0.891; *p* = 0.01638). However, IL-7R rs12516866, TSLP rs2289276, and all TSLPR SNPs did not display any significant genotypic or phenotypic relationship with the CRC cases among either group A or group B (Table 7 and Table 8).

### 3.6. TSLP, TSLPR, and IL-7R Polymorphisms Associations according to the Location of the Tumor in CRC Patients

CRC location may affect colon or rectum or both the sites. For this reason, we evaluated the relation of TSLP, TSLPR, and IL-7R SNPs with CRC development based on the tumor location. Tumor location of CRC is classified as group A for colon location and group B for rectum location. The phenotype and genotype analyses for group A and group B polymorphisms compared to the control are presented in Table 9 and Table 10, respectively. This study observed that *rs10043985* SNP of the *TSLP* gene showed a statistically significant association in the genotypic and phenotypic levels with CRC for both colon and rectum groups. The genotype frequencies in TSLP rs10043985 SNP were as follows; 93% AA in group (A) and 7% AC in controls, while 44% AA and 56% AC in cases. The heterozygous allele AC represents about 17-fold higher correlation with cases compared to the homozygous allele AA (OR: 17.989; CI: 7.185–45.044; *p* < 0.0001). The combinations of AC+CC genotypes were about 17-fold higher in cases, compared to the wild type genotype (OR: 17.989; CI: 7.185–45.044; *p* < 0.0001). In addition, we also found significant phenotypic association with cases. The frequency of the phenotype was 97% A and 3% C in control and 72% A and 28% C in cases, about 11.425-fold increased association of the C phenotype with cases, compared to the phenotype A (OR: 11.425; CI: 4.879–26.754; *p* < 0.0001). Nevertheless, the genetic frequency of group (B) for rs1004395 was: 93% AA and 7% AC in controls while 58% AA and 42% AC in cases. The heterozygous allele AC represents about 10-fold higher correlation with cases compared to the homozygous allele AA (OR: 10.204; CI: 3.705–28.101; *p* < 0.0001). The combinations of AC+CC genotypes were about 10-fold higher in cases (OR: 10.204; CI: 3.705–28.101; *p* < 0.0001). Moreover, this study also found significant phenotypic association with cases. The frequency of the phenotype was 97% A and 3% C in control and 79% A and 21% C in cases, giving 7.782-fold increased correlation of the C phenotype with cases, compared to the phenotype A (OR 7.782; CI: 3.028–19.998; *p* < 0.0001). However, neither of IL-7R SNPs nor TSLPR SNPs showed any significant phenotypic or genotypic association with cases in A group and B group (Table 9 and Table 10).

## 4. Discussion

CRC is a major cause of cancer-related morbidity, mortality, and human health problems worldwide [1]. It was reported that CRC can occur by the interaction between various genetic factors and environmental factors. The association between inflammation and CRC development was described many years ago [13,14,15]. Additionally, the expression levels of multiple genes were found to be down-regulated among CRC patients [44,45]. It was shown that the alterations in the expression levels might be caused by the presence of polymorphisms in different regions of the gene, particularly, either in regulatory, promoter, or exon regions [46,47,48].

A number of studies showed associations between TSLP, TSLPR, and IL-7R expression levels and polymorphisms with various diseases [25,49]. These genes are involved in the innate immune system, and hence, changes in expression levels might cause abnormal expressions at the molecular and protein level. Alterations in the functions of the immune system lead to different types of diseases, for example, in the case of asthma [50,51]. In fact, no known studies have been reported regarding the association of mutations in these cytokines with CRC. In this study, with the help of a large CRC patient group, we investigated the correlation between TSLP, TSLPR, and IL-7R polymorphisms and their expression levels in CRC, which were previously not reported among Saudi population. Therefore, we have selected seven SNPs; rs10043985 (located in the promoter region) and rs2289276 (located in 5′-UTR region) in TSLP; rs36139698, rs36177645, rs36133495 in exon region of TSLPR; and rs1053496 and rs12516866 in promoter of IL-7R to investigate the association between polymorphisms/expression levels of these genes and CRC risk in Saudi population. The choice of the SNP is based on it localization in regulatory region of the gene and on it association with other diseases according to literature review. All TSLPR SNPs are located in the exon region, while IL-7R SNP rs12516866 is located in the promoter region and IL-7R SNP rs1053496 is located in the 3′-UTR region.

We found that TSLP rs10043985 showed a strong significant correlation with CRC in all the parameters tested among Saudi patients, which inferred that this mutation in the promoter region of *TSLP* gene might play a detrimental role in CRC through its dysregulation of the gene expression. This study also supports other our previously work on the same SNP among cigarette smokers [43], and in other diseases, such as, breast cancer [40]. Hence, TSLP SNP rs10043985 SNP might be used as biomarker for CRC. However, another SNP of *TSLP* gene, *rs2289276*, did not show any significant result with CRC during the comparison with all clinical parameters, indicating that this SNP does not have a role in CRC development as also indicated by other studies [40,43]. In addition, we have identified an association between IL-7R rs1053496 SNP with CRC but only in some of the examined parameters. IL-7R rs1053496 SNP displayed a protective role within CRC, like in atopic dermatitis [52]. This association was found only in male subjects both in the phenotypic and genotypic levels, and also in CRC patients who are under 57 years of age in the phenotypic level. On the other hand, IL-7R rs1053496 SNP showed no association with CRC in female subjects or in CRC patients who are more than 57 years of age. Surprisingly, none of the TSLPR variants showed any correlation with CRC. However, in other studies, at least one of the TSLPR SNPs, rs36139698, rs36177645, and rs36133495, showed an association with the disease [50]. In our study, these SNPs of TSLPR played no significant role in colorectal carcinogenesis, so these SNPs cannot be classified as biomarkers for CRC. 

TSLP plays a role in B-cell and T-cell development and proliferation and is also involved in inflammatory responses. TSLP binds to the complex containing TSLPR and IL-7R, providing a high affinity binding which induces the signaling by JAK1, JAK2, and STAT proteins. TSLP, along with TSLPR, plays a dual role in tumor progression, either inducing or preventing the tumor formation depending on the tumor type [31]. In our study, TSLP and TSLPR mRNAs were significantly overexpressed and IHC also showed a similar increase in the levels of expressed proteins. These results correlated well with previous studies that showed the expression levels of TSLP in breast cancer patients [40,53]. IL-7R-α subunit expressions (mRNA and protein levels) were increased in colon cancer tissues compared to matching normal tissues. This explains an associated with Il-7R expression and colon cancer development. In addition, our results on lovo cells (colon cancer cells) shown that, TSLP is able to increase the expression of same cytokines inflammatory as Il-1, Il-4, Il-12 and IL-23 and in turn, these cytokines increase TSLP expression more than 4 times (results not shown). The increase of TSLP and TSLP receptors expression is closely associated with an increase of inflammation.

## 5. Conclusions

This is the first known study to examine the correlation between the expression and genetic variants of TSLP, TSLPR, IL-7R genes with CRC progression in Saudi population. This paper showed that TSLP rs10043985 polymorphism and TSLP/TSLPR expression are associated with an increased risk of CRC in Saudi populations, and therefore they can be used as potential biomarkers for CRC development and diagnosis. Lastly, additional investigations are required on a larger number of patients from diverse ethnicities to confirm the genetic association of these variants to CRC. 

## Figures and Tables

**Figure 1 genes-12-01386-f001:**
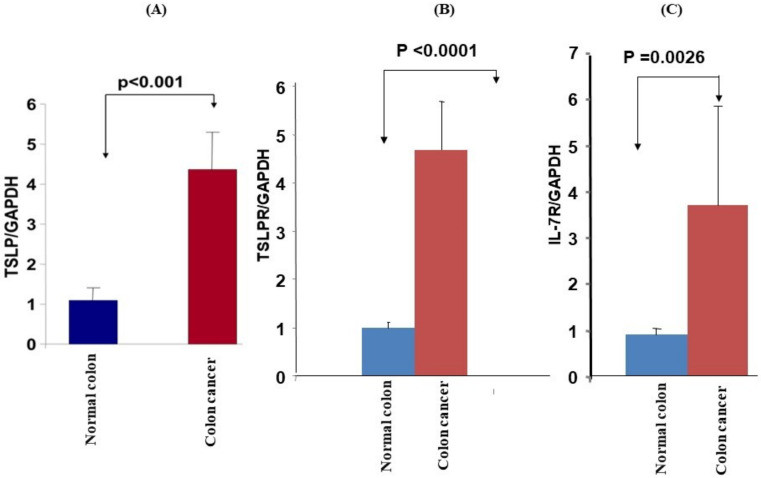
qRT-PCR analyses of TSLP, TSLPR, and IL-7R expressions in matching normal and colon cancer tissues. mRNA expressions of (**A**) TSLP, (**B**) TSLPR, and (**C**) IL-7R in colon cancer tissues when compared to normal colon tissues. The expression levels were normalized to GAPDH reference gene.

**Figure 2 genes-12-01386-f002:**
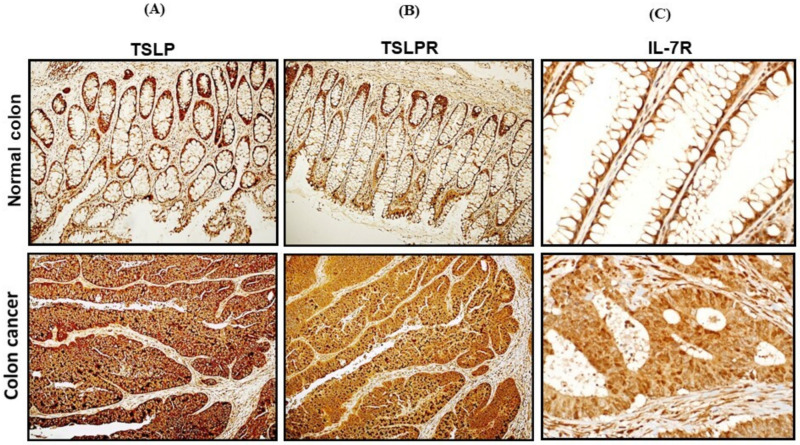
IHC array analyses for TSLP and TSLPR-α, and IL-7R-α in matching normal and colon cancer tissues. Protein levels of (**A**) TSLP, (**B**) TSLPR-α, and (**C**) IL-7R-α in colon cancer tissues compared to normal colon tissues by using a specific antibody for each gene. (**D**) Summary of TSLP, TSLPR and IL-7R expression on normal and colon cancer tissues. Positive cells in the tissues were estimated as follows: Positive staining was estimated as follows: no positive color (0 points), <20% positive staining (1 point), 21–50% positive staining (2 points), 51–75% positive staining (3 points), >75% positive staining (4 points). *** *p* < 0.0005.

**Table 1 genes-12-01386-t001:** Primers used for quantitative RT–PCR.

Gene	Forward Primer (5′-3′)	Reverse Primer (5′-3′)
*TSLP*	TATGAGTGGGACCAAAAGTACCG	GGGATTGAAGGTTAGGCT CTGG
*TSLPR*	GAGTGGCAGTCCAAACAGGAA	ACATCCTCCATAGCCTTCACC
*IL-7R*	TGGACGCATGTGAATTTATC	CATTCACTCCAGAAGCCTTT
*GAPDH*	GGTATCGTCGAAGGACTCATGAC	ATGCCAGTGAGCTTCCCGTTCAGC

**Table 2 genes-12-01386-t002:** TSLP, TSLPR, and IL-7R SNP numbers and characteristics.

Gene	SNP ID	Base Change	Position
** *TSLP* **	rs10043985	A/C	Promoter
	rs2289276	C/T	Intron Variant
** *TSLPR* **	rs36139698	C/T	Exon
	rs36177645	A/G	Exon
	rs36133495	C/T	Exon
** *IL-7R* **	rs12516866	G/T	Intron Variant
	rs1053496	C/T	Downstream Non Coding Transcript Variant

**Table 3 genes-12-01386-t003:** Clinical Characteristic of the study patients.

	Characteristic	CRC Case	Control
Samples No. (%)	---	112 (100%)	108 (100%)
Gender	Male	64 (57.14%)	61 (56.48%)
	Female	48 (42.86%)	47 (43.52%)
Age	Above 57	47 (41.96%)	56 (42.59%)
	Below 57	65 (58.04%)	62 (57.41%)
Tumor Localization	Colon	61 (54.46%)	---
	Rectum	51 (45.54%)	---

**Table 4 genes-12-01386-t004:** Genotype frequencies of TSLP, TSLPR, and IL-7R gene polymorphisms in CRC and controls.

Gene	SNP ID	Genotype	Cases	Controls	OR	(95% CI)	χ^2^-Value	*p*-Value *
** *TSLP* **	**rs10043985**	AA	51 (0.46)	100 (0.93)	Ref			
		AC	59 (0.54)	7 (0.07)	16.52	7.042–38.783	56.84	<0.0001 *
		CC	0 (0)	0 (0)				
		AC+CC	59 (0.54)	7 (0.07)	16.527	7.042–38.783	56.84	<0.0001 *
		A	161 (0.73)	207 (0.97)	Ref			
		C	59 (0.27)	7 (0.03)	10.837	4.820–24.363	46.65	<0.0001 *
	**rs2289276**	CC	50 (0.49)	47 (0.44)	Ref			
		CT	41 (0.40)	51 (0.48)	0.756	0.426–1.339	0.92	0.33702
		TT	12 (0.11)	9 (0.08)	1.253	0.484–3.246	0.22	0.64147
		CT+TT	53 (0.51)	60 (0.56)	0.830	0.482–1.429	0.45	0.50215
		C	141 (0.68)	145 (0.68)	Ref			
		T	65 (0.32)	69 (0.32)	0.969	0.643–1.460	0.02	0.87952
** *TSLPR* **	**rs36133495**	CC	15 (0.14)	16 (0.14)	Ref			
		CT	47 (0.46)	53 (0.49)	0.946	0.422–2.119	0.02	0.89250
		TT	41 (0.40)	40 (0.37)	1.093	0.478–2.503	0.04	0.83273
		CT+TT	88 (0.86)	93 (0.86)	1.009	0.471–2.163	0.00	0.98097
		C	77 (0.37)	85 (0.39)	Ref			
		T	129 (0.63)	133 (0.61)	1.071	0.723–1.585	0.12	0.73275
	**rs36177645**	AA	6 (0.06)	5 (0.06)	Ref			
		AG	42 (0.42)	41 (0.46)	0.854	0.242–3.017	0.06	0.80582
		GG	52 (0.52)	43 (0.48)	1.008	0.288–3.53	0.00	0.99037
		AG+GG	94 (0.94)	84 (0.94)	0.933	0.275–3.167	0.01	0.91085
		A	54 (0.27)	51 (0.29)	Ref			
		G	146 (0.63)	127 (0.71)	1.086	0.692–1.704	0.13	0.72044
	**rs36139698**	CC	4 (0.04)	4 (0.04)	Ref			
		CT	28 (0.26)	37 (0.34)	0.757	0.174–3.292	0.14	0.70960
		TT	76 (0.70)	69 (0.62)	1.101	0.265–4.574	0.02	0.89414
		CT+TT	104 (0.96)	106 (0.96)	0.981	0.239–4.027	0.00	0.97891
		C	36 (0.17)	45 (0.20)	Ref			
		T	180 (0.83)	175 (0.80)	1.286	0.791–2.089	1.03	0.30926
** *IL-7R* **	**rs1053496**	CC	33 (0.30)	17 (0.18)	Ref			
		CT	38 (0.35)	37 (0.44)	0.529	0.252–1.109	2.87	0.08999
		TT	39 (0.35)	43 (0.38)	0.467	0.226–0.968	4.26	0.03903 *
		CT+TT	77 (0.70)	80 (0.82)	0.496	0.255–0.963	4.38	0.03640 *
		C	104 (0.47)	71 (0.37)	Ref			
		T	116 (0.53)	123 (0.63)	0.644	0.434–0.955	4.8	0.02823 *
	**rs12516866**	GG	47 (0.43)	50 (0.46)	Ref			
		GT	55 (0.50)	44 (0.40)	1.330	0.758–2.332	0.99	0.31971
		TT	8 (0.07)	15 (0.14)	0.567	0.220–1.461	1.40	0.23679
		GT+TT	63 (0.57)	59 (0.54)	1.136	0.666–1.937	0.22	0.63952
		G	149 (0.68)	144 (0.66)	Ref			
		T	71 (0.32)	74 (0.34)	0.927	0.623–1.381	0.14	0.71001

Notes: * represent significant results (*p* < 0.05); Ref = Reference allele.

**Table 5 genes-12-01386-t005:** Genotype frequencies of TSLP, TSLPR, and IL-7R gene polymorphisms in CRC and controls in individuals below 57 years of old.

Gene	SNP ID	Genotype	Cases	Controls	OR	(95% CI)	χ^2^-Value	*p*-Value *
** *TSLP* **	**rs10043985**	AA	25 (0.46)	59 (0.94)	Ref			
		AC	29 (0.54)	4 (0.06)	17.11	5.443–53.781	32.20	<0.0001 *
		CC	0 (0)	0 (0)				
		AC+CC	29 (0.54)	4 (0.06)	17.110	5.443–53.781	32.20	<0.0001 *
		A	79 (0.73)	122 (0.97)	Ref			
		C	29 (0.27)	4 (0.03)	11.196	3.79–33.069	26.91	<0.0001 *
	**rs2289276**	CC	22 (0.44)	29 (0.47)	Ref			
		CT	19 (0.38)	26 (0.43)	0.963	0.428–2.167	0.01	0.92793
		TT	9 (0.18)	6 (0.1)	1.977	0.612–6.385	1.32	0.25001
		CT+TT	28 (0.56)	32 (0.53)	1.153	0.544–2.445	0.14	0.70955
		C	63 (0.63)	84 (0.69)	Ref			
		T	37 (0.37)	38 (0.31)	1.298	0.743–2.269	0.84	0.35899
** *TSLPR* **	**rs36139698**	CC	6 (0.12)	11 (0.17)	Ref			
		CT	25 (0.50)	33 (0.51)	1.389	0.452–4.266	0.33	0.56528
		TT	19 (0.38)	21 (0.32)	1.659	0.514–5.357	0.72	0.39555
		CT+TT	44 (0.88)	54 (0.83)	1.494	0.512–4.361	0.54	0.46089
		C	37 (0.37)	55 (0.42)	Ref			
		T	63 (0.63)	75 (0.58)	1.249	0.732–2.131	0.66	0.41534
	**rs36177645**	AA	2 (0.04)	5 (0.1)	Ref			
		AG	23 (0.47)	25 (0.45)	2.300	0.406–13.037	0.92	0.33692
		GG	24 (0.49)	25 (0.45)	2.400	0.424–13.576	1.03	0.31118
		AG+GG	47 (0.96)	50 (0.90)	2.350	0.435–12.704	1.04	0.30880
		A	27 (0.28)	35 (0.32)	Ref			
		G	71 (0.72)	75 (0.68)	1.227	0.675–2.231	0.45	0.50184
	**rs36133495**	CC	2 (0.04)	4 (0.06)	Ref			
		CT	15 (0.28)	22 (0.34)	1.364	0.221–8.415	0.11	0.73767
		TT	37 (0.68)	39 (0.60)	1.897	0.328–10.984	0.53	0.46854
		CT+TT	52 (0.96)	61 (0.94)	1.705	0.300–9.687	0.37	0.54309
		C	19 (0.18)	30 (0.23)	Ref			
		T	89 (0.82)	100 (0.77)	1.405	0.740–2.670	1.09	0.29752
** *IL-7R* **	**rs1053496**	CC	16 (0.30)	9 (0.15)	Ref			
		CT	18 (0.33)	22 (0.37)	0.460	0.165–1.285	2.23	0.13568
		TT	20 (0.37)	29 (0.48)	0.388	0.143–1.050	3.56	0.05913
		CT+TT	38 (0.70)	51 (0.85)	0.419	0.167–1.050	3.55	0.05944
		C	50 (0.46)	40 (0.33)	Ref			
		T	58 (0.54)	80 (0.67)	0.580	0.339–0.991	4.00	0.04556 *
	**rs12516866**	GG	23 (0.43)	31 (0.48)	Ref			
		GT	28 (0.52)	24 (0.38)	1.572	0.730–3.386	1.34	0.24636
		TT	3 (0.05)	9 (0.14)	0.449	0.109–1.847	1.27	0.25925
		GT+TT	31 (0.57)	33 (0.52)	1.266	0.611–2.624	0.40	0.52548
		G	74 (0.69)	86 (0.67)	Ref			
		T	34 (0.31)	42 (0.33)	0.941	0.544–1.628	0.05	0.82742

Notes: * represent significant results (*p* < 0.05); Ref = Reference allele.

**Table 6 genes-12-01386-t006:** Genotype frequencies of TSLP, TSLPR, and IL-7R gene polymorphisms CRC and controls in individuals above 57 years of old.

Gene	SNP ID	Genotype	Cases	Controls	OR	(95% CI)	χ^2^-Value	*p*-Value *
** *TSLP* **	**rs10043985**	AA	26 (0.46)	41 (0.93)	Ref			
		AC	30 (0.54)	3 (0.07)	15.769	4.365–56.973	24.36	<0.0001 *
		CC	0 (0)	0 (0)				
		AC+CC	30 (0.54)	3 (0.07)	15.769	4.365–56.973	24.36	<0.0001 *
		A	82 (0.73)	85 (0.97)	Ref			
		C	30 (0.27)	3 (0.03)	10.366	3.045–35.286	19.55	<0.0001 *
	**rs2289276**	CC	28 (0.53)	18 (0.4)	Ref			
		CT	22 (0.41)	25 (0.54)	0.566	0.248–1.290	1.85	0.17390
		TT	3 (0.06)	3 (0.06)	0.643	0.117–3.541	0.26	0.60980
		CT+TT	25 (0.47)	28 (0.60)	0.574	0.258–1.279	1.86	0.17285
		C	78 (0.74)	61 (0.66)	Ref			
		T	28 (0.26)	31 (0.34)	0.706	0.383–1.301	1.25	0.26393
** *TSLPR* **	**rs36139698**	CC	9 (0.17)	5 (0.11)	Ref			
		CT	22 (0.41)	20 (0.46)	0.611	0.175–2.132	0.60	0.43776
		TT	22 (0.41)	19 (0.43)	0.643	0.184–2.254	0.48	0.48877
		CT+TT	44 (0.83)	39 (0.89)	0.627	0.194–2.030	0.61	0.43317
		C	40 (0.38)	30 (0.34)	Ref			
		T	66 (0.62)	58 (0.66)	0.853	0.473–1.540	0.28	0.59869
	**rs36177645**	AA	4 (0.08)	0 (0.00)	Ref			
		AG	19 (0.37)	16 (0.47)	0.131	0.007–2.623	3.10	0.07826
		GG	28 (0.55)	18 (0.53)	0.171	0.009–3.369	2.45	0.11785
		AG+GG	47 (0.92)	34 (1.00)	0.153	0.008–2.936	2.80	0.09436
		A	27 (0.26)	16 (0.24)	Ref			
		G	75 (0.74)	52 (0.76)	0.855	0.419–1.743	0.19	0.66561
	**rs36133495**	CC	2 (0.04)	0 (0.00)	Ref			
		CT	13 (0.24)	15 (0.33)	0.174	0.008–3.956	2.14	0.14323
		TT	39 (0.72)	30 (0.67)	0.259	0.012–5.596	1.51	0.21978
		CT+TT	52 (0.96)	45 (1.0)	0.231	0.011–4.933	1.70	0.19215
		C	17 (0.16)	15 (0.17)	Ref			
		T	91 (0.84)	75 (0.83)	1.071	0.501–2.286	0.03	0.86010
** *IL-7R* **	**rs1053496**	CC	17 (0.30)	8 (0.22)	Ref			
		CT	20 (0.36)	15 (0.41)	0.627	0.214–1.837	0.73	0.39379
		TT	19 (0.34)	14 (0.37)	0.639	0.215–1.895	0.66	0.41779
		CT+TT	39 (0.70)	29 (0.78)	0.633	0.240–1.666	0.86	0.35235
		C	54 (0.48)	31 (0.42)	Ref			
		T	58 (0.52)	43 (0.58)	0.774	0.428–1.400	0.72	0.39688
	**rs12516866**	GG	24 (0.42)	19 (0.42)	Ref			
		GT	27 (0.48)	20 (0.44)	1.069	0.464–2.462	0.02	0.87592
		TT	5 (0.09)	6 (0.13)	0.660	0.174–2.496	0.38	0.53863
		GT+TT	32 (0.57)	26 (0.57)	0.974	0.441–2.155	0.00	0.94886
		G	75 (0.67)	58 (0.64)	Ref			
		T	37 (0.33)	32 (0.36)	0.894	0.499–1.604	0.14	0.7074

Notes: * represent significant results (*p* < 0.05); Ref = Reference allele.

**Table 7 genes-12-01386-t007:** Genotype frequencies of TSLP, TSLPR, and IL-7R gene polymorphisms in CRC and controls in male subjects.

Gene	SNP ID	Genotype	Cases	Controls	OR	(95% CI)	χ^2^-Value	*p*-Value *
** *TSLP* **	**rs10043985**	AA	26 (0.42)	55 (0.93)	Ref			
		AC	36 (0.58)	4 (0.07)	19.038	6.129–59.139	35.93	<0.0001 *
		CC	0 (0)	0 (0)				
		AC+CC	36 (0.58)	4 (0.07)	19.038	6.129–59.139	35.93	<0.0001 *
		A	88 (0.71)	114 (0.97)	Ref			
		C	36 (0.29)	4 (0.03)	11.659	4.000–33.983	28.82	<0.0001 *
	**rs2289276**	CC	30 (0.51)	26 (0.45)	Ref			
		CT	25 (0.42)	27 (0.46)	0.802	0.377–1.709	0.33	0.56819
		TT	4 (0.07)	5 (0.09)	0.693	0.168–2.856	0.26	0.61087
		CT+TT	29 (0.49)	32 (0.55)	0.785	0.380–1.625	0.42	0.51458
		C	85 (0.72)	79 (0.68)	Ref			
		T	33 (0.28)	37 (0.32)	0.829	0.473–1.452	0.43	0.51149
** *TSLPR* **	**rs36139698**	CC	10 (0.17)	10 (0.17)	Ref			
		CT	29 (0.48)	31 (0.52)	0.935	0.340–2.574	0.02	0.89725
		TT	21 (0.35)	19 (0.31)	1.105	0.378–3.235	0.03	0.85505
		CT+TT	50 (0.83)	50 (0.83)	1.000	0.383–2.612	0.00	1.000
		C	49 (0.41)	51 (0.43)	Ref			
		T	71 (0.59)	69 (0.57)	1.071	0.641–1.789	0.07	0.79343
	**rs36177645**	AA	3 (0.05)	4 (0.08)	Ref			
		AG	27 (0.5)	19 (0.39)	1.895	0.380–9.459	0.62	0.43088
		GG	25 (0.45)	26 (0.53)	1.282	0.260–6.315	0.09	0.75964
		AG+GG	52 (0.55)	45 (0.92)	1.541	0.327–7.254	0.30	0.58209
		A	33 (0.30)	27 (0.28)	Ref			
		G	77 (0.70)	71 (0.72)	0.887	0.486–1.620	0.15	0.69716
	**rs36133495**	CC	2 (0.03)	3 (0.05)	Ref			
		CT	19 (0.31)	17 (0.28)	1.676	0.249–11.266	0.29	0.59222
		TT	40 (0.66)	41 (0.67)	1.463	0.232–9.228	0.17	0.68376
		CT+TT	59 (0.97)	58 (0.95)	1.526	0.246–9.470	0.21	0.64791
		C	23 (0.19)	23 (0.19)	Ref			
		T	99 (0.81)	99 (0.81)	1.000	0.526–1.900	0.00	1.00
** *IL-7R* **	**rs1053496**	CC	22 (0.35)	9 (0.17)	Ref			
		CT	20 (0.32)	20 (0.37)	0.409	0.152–1.104	3.18	0.07464
		TT	21 (0.33)	25 (0.46)	0.344	0.130–0.905	4.81	0.02824 *
		CT+TT	41 (0.65)	45 (0.63)	0.373	0.154–0.902	4.97	0.02572 *
		C	64 (0.51)	38 (0.35)	Ref			
		T	62 (0.49)	70 (0.65)	0.526	0.310–0.891	5.76	0.01638 *
	**rs12516866**	GG	27 (0.43)	29 (0.48)	Ref			
		GT	31 (0.49)	22 (0.37)	1.513	0.710–3.227	1.15	0.28251
		TT	5 (0.08)	9 (0.15)	0.597	0.178–2.006	0.71	0.40105
		GT+TT	36 (0.57)	31 (0.52)	1.247	0.613–2.539	0.37	0.54213
		G	85 (0.67)	80 (0.67)	Ref			
		T	41 (0.33)	40 (0.33)	0.965	0.567–1.642	0.02	0.89467

Notes: * represent significant results (*p* < 0.05); Ref = Reference allele.

**Table 8 genes-12-01386-t008:** Genotype frequencies of TSLP, TSLPR, and IL-7R gene polymorphisms in CRC and controls in female subjects.

Gene	SNP ID	Genotype	Cases	Controls	OR	(95% CI)	χ^2^-Value	*p*-Value *
** *TSLP* **	**rs10043985**	AA	25 (0.52)	43 (0.93)	Ref			
		AC	23 (0.48)	3 (0.07)	13.187	3.593–48.395	20.12	<0.0001 *
		CC	0 (0)	0 (0)				
		AC+CC	23 (0.48)	3 (0.07)	13.187	3.593–48.395	20.12	<0.0001 *
		A	73 (0.76)	89 (0.97)	Ref			
		C	23 (0.24)	3 (0.03)	9.347	2.699–32.374	16.89	<0.0001 *
	**rs2289276**	CC	20 (0.46)	20 (0.43)	Ref			
		CT	16 (0.36)	24 (0.51)	0.667	0.275–1.616	0.81	0.36869
		TT	8 (0.18)	3 (0.06)	2.667	0.617–11.535	1.80	0.17973
		CT+TT	24 (0.54)	27 (0.57)	0.889	0.388–2.036	0.08	0.78050
		C	56 (0.64)	64 (0.68)	Ref			
		T	32 (0.36)	30 (0.32)	1.219	0.660–2.252	0.40	0.52684
** *TSLPR* **	**rs36139698**	CC	5 (0.11)	6 (0.13)	Ref			
		CT	18 (0.42)	22 (0.47)	0.982	0.257–3.751	0.00	0.97859
		TT	20 (0.47)	19 (0.40)	1.263	0.330–4.837	0.12	0.73281
		CT+TT	38 (0.89)	41 (0.87)	1.112	0.314–3.945	0.03	0.86922
		C	28 (0.33)	34 (0.36)	Ref			
		T	58 (0.67)	60 (0.64)	1.174	0.633–2.175	0.26	0.61046
	**Rs36177645**	AA	3 (0.07)	1 (0.02)	Ref			
		AG	15 (0.33)	22 (0.58)	0.227	0.022–2.398	1.74	0.18708
		GG	27 (0.60)	15 (0.40)	0.600	0.057–6.288	0.18	0.66726
		AG+GG	42 (0.93)	37 (0.98)	0.378	0.038–3.796	0.73	0.39246
		A	21 (0.04)	24 (0.31)	Ref			
		G	69 (0.2)	52 (0.68)	1.516	0.763–3.016	1.42	0.23377
	**Rs36133495**	CC	2 (0.04)	1 (0.02)	Ref			
		CT	9 (0.2)	20 (0.43)	0.225	0.018–2.813	1.53	0.21609
		TT	36 (0.76)	26 (0.55)	0.692	0.060–8.046	0.09	0.76777
		CT+TT	45 (0.96)	46 (0.98)	0.489	0.043–5.586	0.34	0.55734
		C	13 (0.14)	22 (0.23)	Ref			
		T	81 (0.86)	72 (0.77)	1.904	0.894–4.053	2.84	0.09173
** *IL-7R* **	**rs1053496**	CC	11 (0.23)	7 (0.17)	Ref			
		CT	18 (0.38)	17 (0.41)	0.674	0.212–2.142	0.45	0.50245
		TT	18 (0.38)	17 (0.41)	0.674	0.212–2.142	0.45	0.50245
		CT+TT	36 (0.76)	34 (0.82)	0.674	0.234–1.939	0.54	0.46266
		C	40 (0.43)	31 (0.38)	Ref			
		T	54 (0.57)	51 (0.62)	0.821	0.448–1.503	0.41	0.52182
	**rs12516866**	GG	20 (0.43)	20 (0.43)	Ref			
		GT	24 (0.51)	22 (0.46)	1.091	0.467–2.547	0.04	0.84057
		TT	3 (0.06)	5 (0.1)	0.600	0.126–2.855	0.42	0.51824
		GT+TT	27 (0.57)	27 (0.56)	1.000	0.441–2.265	0.00	1.000
		G	64 (0.68)	62 (0.66)	Ref			
		T	30 (0.32)	32 (0.34)	0.908	0.494–1.669	0.10	0.75636

Notes: * represent significant results (*p* < 0.05); Ref = Reference allele.

**Table 9 genes-12-01386-t009:** Genotype frequencies of TSLP, TSLPR, and IL-7R gene polymorphisms in CRC and controls in colon tumors.

Gene	SNP ID	Genotype	Cases	Controls	OR	(95% CI)	χ^2^-Value	*p*-Value *
** *TSLP* **	**rs10043985**	AA	27 (0.44)	100 (0.93)	Ref			
		AC	34 (0.56)	7 (0.07)	17.989	7.185–45.044	50.97	<0.0001 *
		CC	0 (0)	0 (0)				
		AC+CC	34 (0.56)	7 (0.07)	17.989	7.185–45.044	50.97	<0.0001 *
		A	88 (0.72)	207 (0.97)	Ref			
		C	34 (0.28)	7 (0.03)	11.425	4.879–26.754	43.88	<0.0001 *
	**rs2289276**	CC	30 (0.50)	47 (0.44)	Ref			
		CT	23 (0.38)	51 (0.48)	0.707	0.361–1.384	1.03	0.31049
		TT	7 (0.12)	9 (0.08)	1.219	0.410–3.620	0.13	0.72175
		CT+TT	30 (0.50)	60 (0.56)	0.783	0.416–1.477	0.57	0.44989
		C	83 (0.69)	145 (0.68)	Ref			
		T	37 (0.31)	69 (0.32)	0.937	0.579–1.517	0.07	0.79058
** *TSLPR* **	**rs36139698**	CC	7 (0.12)	16 (0.14)	Ref			
		CT	31 (0.53)	53 (0.49)	1.337	0.495–3.607	0.33	0.56564
		TT	20 (0.35)	40 (0.37)	1.143	0.405–3.226	0.06	0.80082
		CT+TT	51 (0.88)	93 (0.86)	1.253	0.484–3.246	0.22	0.64123
		C	45 (0.39)	85 (0.39)	Ref			
		T	71 (0.61)	133 (0.61)	1.008	0.635–1.601	0.00	0.97185
	**rs36177645**	AA	3 (0.05)	5 (0.06)	Ref			
		AG	26 (0.46)	41 (0.46)	1.057	0.233–4.800	0.01	0.94285
		GG	27 (0.48)	43 (0.48)	1.047	0.231–4.738	0.00	0.95294
		AG+GG	53 (0.94)	84 (0.94)	1.052	0.241–4.583	0.00	0.94660
		A	32 (0.29)	51 (0.29)	Ref			
		G	80 (0.71)	127 (0.71)	1.004	0.595–1.694	0.00	0.98825
	**rs36133495**	CC	2 (0.03)	4 (0.04)	Ref			
		CT	17 (0.28)	37 (0.34)	0.919	0.153–5.514	0.01	0.92629
		TT	42 (0.69)	69 (0.62)	1.217	0.214–6.937	0.05	0.82442
		CT+TT	59 (0.97)	106 (0.96)	1.113	0.198–6.26	0.01	0.90308
		C	21 (0.17)	45 (0.20)	Ref			
		T	101 (0.83)	175 (0.80)	1.237	0.697–2.193	0.53	0.46684
** *IL-7R* **	**rs1053496**	CC	16 (0.26)	17 (0.18)	Ref			
		CT	23 (0.38)	37 (0.44)	0.660	0.280–1.558	0.90	0.34250
		TT	22 (0.36)	43 (0.38)	0.544	0.231–1.277	1.98	0.15984
		CT+TT	0 (0.74)	80 (0.82)	0.598	0.276–1.296	1.72	0.19009
		C	55 (0.45)	71 (0.37)	Ref			
		T	67 (0.55)	123 (0.63)	0.703	0.443–1.115	2.25	0.13373
	**rs12516866**	GG	23 (0.38)	50 (0.46)	Ref			
		GT	33 (0.54)	44 (0.40)	1.630	0.835–3.183	2.06	0.15086
		TT	5 (0.08)	15 (0.14)	0.725	0.235–2.235	0.32	0.57410
		GT+TT	38 (0.62)	59 (0.54)	1.400	0.738–2.656	1.06	0.30216
		G	79 (0.65)	144 (0.66)	Ref			
		T	43 (0.35)	74 (0.34)	1.059	0.665–1.687	0.06	0.80863

Notes: * represent significant results (*p* < 0.05); Ref = Reference allele.

**Table 10 genes-12-01386-t010:** Genotype frequencies of TSLP, TSLPR, and IL-7R gene polymorphisms in CRC and controls in rectal tumors.

Gene	SNP ID	Genotype	Cases	Controls	OR	(95% CI)	χ^2^-Value	*p*-Value *
** *TSLP* **	**rs10043985**	AA	21 (0.58)	100 (0.93)	Ref			
		AC	15 (0.42)	7 (0.07)	10.204	3.705–28.101	25.53	<0.0001 *
		CC	0 (0)	0 (0)				
		AC+CC	15 (0.42)	7 (0.07)	10.204	3.705–28.101	25.53	<0.0001 *
		A	57 (0.79)	207 (0.97)	Ref			
		C	15 (0.21)	7 (0.33)	7.782	3.028–19.998	23.40	<0.0001 *
	**rs2289276**	CC	14 (0.42)	47 (0.44)	Ref			
		CT	14 (0.42)	51 (0.48)	0.922	0.398–2.135	0.04	0.84886
		TT	5 (0.15)	9 (0.08)	1.865	0.537–6.480	0.98	0.32204
		CT+TT	19 (0.57)	60 (0.56)	1.063	0.483–2.340	0.02	0.87917
		C	42 (0.64)	145 (0.68)	Ref			
		T	24 (0.36)	69 (0.32)	1.201	0.674–2.140	0.39	0.53435
** *TSLPR* **	**rs36139698**	CC	3 (0.1)	16 (0.14)	Ref			
		CT	12 (0.36)	53 (0.49)	1.208	0.303–4.815	0.07	0.78907
		TT	18 (0.55)	40 (0.37)	2.400	0.620–9.284	1.68	0.19533
		CT+TT	30 (91)	93 (0.86)	1.720	0.469–6.313	0.68	0.40874
		C	18 (0.27)	85 (0.39)	Ref			
		T	48 (0.73)	133 (0.61)	1.704	0.930–3.125	3.01	0.08277
	**rs36177645**	AA	2 (0.06)	5 (0.06)	Ref			
		AG	8 (0.26)	41 (0.46)	0.488	0.080–2.970	0.63	0.42879
		GG	21 (0.68)	43 (0.48)	1.221	0.218–6.824	0.05	0.81992
		AG+GG	29 (0.94)	84 (0.94)	0.863	0.159–4.693	0.03	0.86458
		A	12 (0.2)	51 (0.29)	Ref			
		G	50 (0.8)	127 (0.71)	1.673	0.824–3.400	2.05	0.15191
	**rs36133495**	CC	1 (0.03)	4 (0.04)	Ref			
		CT	6 (0.17)	37 (0.34)	0.649	0.062–6.835	0.13	0.71692
		TT	28 (0.8)	69 (0.62)	1.623	0.174–15.169	0.18	0.66823
		CT+TT	34 (0.25)	106 (0.96)	1.283	0.139–11.874	0.05	0.82583
		C	8 (0.11)	45 (0.20)	Ref			
		T	62 (0.89)	175 (0.80)	1.993	0.890–4.461	2.90	0.08877
** *IL-7R* **	**rs1053496**	CC	10 (0.28)	17 (0.18)	Ref			
		CT	12 (0.33)	37 (0.44)	0.551	0.199–1.524	1.33	0.24837
		TT	14 (0.39)	43 (0.38)	0.553	0.206–1.485	1.40	0.23718
		CT+TT	26 (0.72)	80 (0.82)	0.552	0.225–1.356	1.71	0.19156
		C	32 (0.44)	71 (0.37)	Ref			
		T	40 (0.56)	123 (0.63)	0.722	0.417–1.249	1.36	0.24310
	**rs12516866**	GG	16 (0.44)	50 (0.46)	Ref			
		GT	18 (0.50)	44 (0.40)	1.278	0.583–2.805	0.38	0.53976
		TT	2 (0.06)	15 (0.14)	0.417	0.086–2.021	1.24	0.26562
		GT+TT	20 (0.56)	59 (0.54)	1.059	0.497–2.26	0.02	0.88149
		G	50 (0.70)	144 (0.66)	Ref			
		T	22 (0.30)	74 (0.34)	0.856	0.482–1.521	0.28	0.59619

Notes: * represent significant results (*p* < 0.05); Ref = Reference allele.

## Data Availability

All data generated or analyzed during this study are included in this published article.

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
