# Peer review of "Expression and Polymorphism of TSLP/TSLP Receptors as Potential Diagnostic Markers of Colorectal Cancer Progression"

_genes, 2021, doi:10.3390/genes12091386_

Round 1
Reviewer 1 Report
This article performs a case-control study to test the association between colorectal cancer and the single nucleotide polymorphisms (SNPs) located in the thymic stromal lymphopoietin gene, TSLP receptor gene, and interleukin- 7 receptor gene. The rs10043985 was found to have a statistically significant association in the genotypic and phenotypic levels with CRC. However, another statement that rs1053496 polymorphisms can be used as a protective factor for CRC susceptibility is not conclusive.
Below are some suggestions and questions about this article.
- In the Method section, the authors mentioned “For analyzing TSLP, TSLPR, and IL-7R expressions, the differences between two groups were assessed by a paired student t-test”. Please clarify where the paired t-tests were applied. A paired t-test is used when we are interested in the difference between two variables for the same subject. Therefore, it is not suitable for this study.
- The expression of “fold change” is incorrect in this article. I just point out some of them for example. Please correct all of them in the articles. For example, in line 229, “The mRNA expressions of TSLP showed an increase of about 5 folds in colon cancer tissues compared to normal colon tissues”, here, the 5 folds should be 4 fold. Similarly, 5 fold higher in line 236 and 11 fold increased in line 282 should be corrected.
- The last three sentences in the discussion make me confused. In the first sentence, “In contrast, IL-7R-α subunit at mRNA and protein levels were increased in colon cancer tissues compared to matching normal tissues.” please be careful about the use of logic expression. For the second sentence, “This explains closely the associated with colon cancer progression and an increase of inflammation caused mainly by Th2 cytokines overexpressions which in turn are responsible for the overexpression of TSLP and this loop between TSLP and Th2 leads to increase the inflammation in colon cancer via hyperactivation of TSLPR pathway.” I cannot figure out how the statement before this sentence explains this viewpoint. Also, what is your point of the last sentence? “However our results on lovo cells (colon cancer cells) shown that, Il-1, Il-4, Il-12 and IL-23 cytokines increases TSLP expression more than 4 times (results not shown).” I cannot see the purpose of this statement.
- The author claimed in the conclusion that “this is the first known study to examine the correlation between the expression and genetic variants of TSLP, TSLPR, IL-7R genes with CRC progression in Saudi population”. Is there any other paper studied the correlation in other population? If yes, you can compare their results with yours.
- Also, in the conclusion, the author mentioned “however, IL-7R rs1053496 polymorphisms can be used as a protective factor for CRC susceptibility”. However, according to the authors’ results, we cannot see the significant difference between control and case groups in the people aged 57+, female, and location of tumor after separating the population into different sub-population. Take gender as an example, in this case, if we pool the two sub-populations, male and female, as a total population to get positive results, we actually result in a false positive association in the female sub-population. Therefore, the author should narrow the population when he makes a conclusion.
Author Response
20/08/2021
Manuscript ID: genes-1326369
Genes
“Expression and polymorphism of TSLP/TSLP Receptors as potential diagnostic markers of colorectal cancer progression”
Reviewers' comments:
Reviewer 1:
This article performs a case-control study to test the association between colorectal cancer and the single nucleotide polymorphisms (SNPs) located in the thymic stromal lymphopoietin gene, TSLP receptor gene, and interleukin- 7 receptor gene. The rs10043985 was found to have a statistically significant association in the genotypic and phenotypic levels with CRC. However, another statement that rs1053496 polymorphisms can be used as a protective factor for CRC susceptibility is not conclusive.
Below are some suggestions and questions about this article.
1- In the Method section, the authors mentioned “For analyzing TSLP, TSLPR, and IL-7R expressions, the differences between two groups were assessed by a paired student t-test”. Please clarify where the paired t-tests were applied. A paired t-test is used when we are interested in the difference between two variables for the same subject. Therefore, it is not suitable for this study.
Response, Dear, all our statistical analysis was done by statistical department in Laval university in Canada, according to our statistical department, they confirm that t-Test can be used, especially, to determine if the means of two sets of data are significantly different from each other. There are different types of t test: The one-sample t test used to compare an observed mean with a theoretical mean. The independent t test (or two sample t test) used to compare the means of two unrelated groups of samples.
All our previous works were done by the same t- test:
1- Gene. 2021 Jan 15; 766:145092. doi:10.1016/j.gene.2020.145092. Correlation between genetic variation in thymine DNA glycosylase and smoking behavior
2- PLoS One. 2021 Jan 22; 16(1):e0245133. doi:10.1371/journal.pone.0245133. Association between tobacco substance usage and a missense mutation in the tumor suppressor gene P53 in the Saudi Arabian population
3- Environ Sci Pollut Res Int. 2021 Jul 13. doi:10.1007/s11356-021-15242-1. The correlation between single nucleotide polymorphisms of the thymic stromal lymphopoietin receptor and breast cancer in a cohort of female patients in Saudi Arabia
4-Saudi J Biol Sci . 2021 Jul; 28(7):3972-3980. doi:10.1016/j.sjbs.2021.04.011. The protective effects of the methylenetetrahydrofolate reductase rs1801131 variant among Saudi smokers
2- The expression of “fold change” is incorrect in this article. I just point out some of them for example. Please correct all of them in the articles. For example, in line 229, “The mRNA expressions of TSLP showed an increase of about 5 folds in colon cancer tissues compared to normal colon tissues”, here, the 5 folds should be 4 fold. Similarly, 5 fold higher in line 236 and 11 fold increased in line 282 should be corrected.
Response: Thank you for your kind suggestion. We have corrected all folds in Lines 26, 235, 242, 288, 323, 325, 359, 398, and 400.
3- The last three sentences in the discussion make me confused. In the first sentence, “In contrast, IL-7R-α subunit at mRNA and protein levels were increased in colon cancer tissues compared to matching normal tissues.” please be careful about the use of logic expression. For the second sentence, “This explains closely the associated with colon cancer progression and an increase of inflammation caused mainly by Th2 cytokines overexpressions which in turn are responsible for the overexpression of TSLP and this loop between TSLP and Th2 leads to increase the inflammation in colon cancer via hyperactivation of TSLPR pathway.” I cannot figure out how the statement before this sentence explains this viewpoint. Also, what is your point of the last sentence? “However our results on lovo cells (colon cancer cells) shown that, Il-1, Il-4, Il-12 and IL-23 cytokines increases TSLP expression more than 4 times (results not shown).” I cannot see the purpose of this statement.
Response: Very nice remark, we have changed this section in discussion part.
4- The author claimed in the conclusion that “this is the first known study to examine the correlation between the expression and genetic variants of TSLP, TSLPR, IL-7R genes with CRC progression in Saudi population”. Is there any other paper studied the correlation in other population? If yes, you can compare their results with yours.
Response: We thank the reviewer for his/her kind suggestion. This is the first work to investigate the association between TSLP polymorphism and colon cancer but we have published this year one article on the association between TSLP/TSLP receptor on breast cancer in the Saudi population (The correlation between single nucleotide polymorphisms of the thymic stromal lymphopoietin receptor and breast cancer in a cohort of female patients in Saudi Arabia. Semlali A, Almutairi MH, Alharbi SN, Alamri AM, Alrefaei AF, Almutairi BO, Rouabhia M. Environ Sci Pollut Res Int. 2021 Jul 13. doi: 10.1007/s11356-021-15242-1). Also, we have not found any paper studied the correlation in other population in the data base.
5- Also, in the conclusion, the author mentioned “however, IL-7R rs1053496 polymorphisms can be used as a protective factor for CRC susceptibility”. However, according to the authors’ results, we cannot see the significant difference between control and case groups in the people aged 57+, female, and location of tumor after separating the population into different sub-population. Take gender as an example, in this case, if we pool the two sub-populations, male and female, as a total population to get positive results, we actually result in a false positive association in the female sub-population. Therefore, the author should narrow the population when he makes a conclusion.
Response: We thank the reviewer for his/her beautiful suggestion. IL-7R rs1053496 polymorphisms can be used as a protective factor for CRC susceptibility has been deleted from the conclusion part.

Reviewer 2 Report
The study analyses the expression of polymorphisms of TSLP genes and their receptors (TSLP-R and IL-7R) as markers for the diagnosis of colorectal cancer.
The authors should explain the rationale for this study, explaining the reason for studying these genes and their relationship to colorectal cancer.
The introduction does not indicate the rationale for carrying out this research in colorectal cancer, nor whether there are previous studies that support it, nor the preclinical data on the expression of these SNPs in colorectal cancer.
It should be detailed how the selection of colorectal cancer cases was done. It is understood that they are samples of tumours diagnosed at early stages, but the tumour stage is not indicated. The conclusions may be influenced by whether the tumours are early or advanced stage tumours. In addition, other histological characteristics are not specified, such as tumour grade, lymphovascular invasion, presence of microsatellite instability, etc.
A very important data for its prognostic and predictive value in terms of treatment is the location. It is only divided into colon and rectum (table 3), while it would be relevant to differentiate between tumours of the right and left colon and rectum.
In terms of tumours, the mean age of the patients was 57 years, younger than usual in other populations. Other factors that may influence the diagnosis of colorectal cancer at an early age (genetic factors such as hereditary syndromes, obesity, sedentary lifestyle, smoking, etc.) should be assessed.
The authors should also indicate more precisely how the controls were selected, and why for each experiment they use a different number of cases and controls.
The discussion should be improved to facilitate understanding of the clinical significance of the results. In addition, lines 458-473 mention results that are not shown, so they should appear before or you should explain what they are refered to.
Author Response
20/08/2021
Manuscript ID: genes-1326369
Genes
“Expression and polymorphism of TSLP/TSLP Receptors as potential diagnostic markers of colorectal cancer progression”
Reviewer 2:
The study analyses the expression of polymorphisms of TSLP genes and their receptors (TSLP-R and IL-7R) as markers for the diagnosis of colorectal cancer.
- The authors should explain the rationale for this study, explaining the reason for studying these genes and their relationship to colorectal cancer.
Response: We thank the reviewer for his/her beautiful suggestion. Colon cancer is considered as inflammatory diseases and one of the novo cytokine inflammatory described by his capacity to induce cancer is TSLP. The variation of TSLP expression can to active TSLP/TSLP Receptors pathways and induce same cytokine inflammatory expression known to induce cancer as Th2 cytokines overexpressions. Also, the variation of TSLP expression is due to genetic modification as the polymorphism for this reason, our hypothesis was the allelic variation of TSLP can to modify the TSLP expression and this variation of TSLP is one of cause for colon cancer progression.
- The introduction does not indicate the rationale for carrying out this research in colorectal cancer, nor whether there are previous studies that support it, nor the preclinical data on the expression of these SNPs in colorectal cancer.
Response: We have added two small sentences in the Introduction section for the choice of SNPs and our previous study reported the link of these SNPs and breast cancer development in the same population.
- It should be detailed how the selection of colorectal cancer cases was done. It is understood that they are samples of tumours diagnosed at early stages, but the tumour stage is not indicated. The conclusions may be influenced by whether the tumours are early or advanced stage tumours. In addition, other histological characteristics are not specified, such as tumour grade, lymphovascular invasion, presence of microsatellite instability, etc.
Response: I agree with the reviewer comment but the problem is the size of sample. In the Saudi population, it was difficult to collect more samples because the country is too withdrawn with a lot of religious and traditional obstacles and person not collaborate for given tissues.
At first, we aimed to compare these SNPs and TSLP expression in different stage of cancer and different histological characteristics. With our small size of samples collected during 4 years, it was not possible to compare between subgroup of clinical data.
- A very important data for its prognostic and predictive value in terms of treatment is the location. It is only divided into colon and rectum (table 3), while it would be relevant to differentiate between tumours of the right and left colon and rectum.
Response: Nice remark, as answer in previous question, the size of sample is obstacle to compare right and left of colon and rectum. After discussion with Dr. Abdulrahman Aljebreen, Dr. Othman alharbi and Dr. Nahla Ali Azzam, my clinicians collaborators who give me the samples, they suggest to compare just colon and rectum and age and sex because statistical not possible due to size of samples.
- In terms of tumours, the mean age of the patients was 57 years, younger than usual in other populations. Other factors that may influence the diagnosis of colorectal cancer at an early age (genetic factors such as hereditary syndromes, obesity, sedentary lifestyle, smoking, etc.) should be assessed.
Response: I agree with you; therefore, in this study we have selected just the patients having cancer non-smoker no diabetic etc. suffering only from the colon cancer
- The authors should also indicate more precisely how the controls were selected, and why for each experiment they use a different number of cases and controls.
Response: For gene expression study, the normal tissues were collected from the same patient having colon cancer; the tissue was analysis by Dr. Maha Arafah our pathologist. However, for genotyping study, the blood were collected from patient suffering only from colon cancer and normal blood were collected from the healthy patient don’t have any disease and no smoking after control examination in King Khaled Hospital. For each patient, we give number to identify in our tissues bank developed in our institution.
- The discussion should be improved to facilitate understanding of the clinical significance of the results. In addition, lines 458-473 mention results that are not shown, so they should appear before or you should explain what they are referred to.
Response: We thank the reviewer for his/her kind suggestion. Some modification were added to the Discussion section according to the reviewer recommendations

Round 2
Reviewer 1 Report
Before publication, some points are still needed to be discussed.
- For the t-test, the statement in the author’s response is right. However, obviously, they didn’t figure out the key point of my last review comments. I did’t said that the t-test cannot be used for comparing two means. The key point is the “paired t-test”, whose definition is easy to be found by google. According to the authors’ response, absolutely, “paired t-test” is not a correct method. So, please contact your collaborators to figure out which method they used.
- “In contrast, IL-7R-α subunit at mRNA and protein levels were increased in colon cancer tissues compared to matching normal tissues.” At here, I don’t think “in contrast” is a correct expression. If one thing is in contrast to another, it is very different from it.
- “The increase of TSLP and TSLP receptors expression is closely associated with an increase of inflammation” The author should put this sentence after the last sentence. This is because the statement and evidence before the last sentence cannot deduce the association between TSLP and the increase of inflammation.
Author Response
- For the t-test, the statement in the author’s response is right. However, obviously, they didn’t figure out the key point of my last review comments. I did’t said that the t-test cannot be used for comparing two means. The key point is the “paired t-test”, whose definition is easy to be found by google. According to the authors’ response, absolutely, “paired t-test” is not a correct method. So, please contact your collaborators to figure out which method they used.
Reply: Thank you. We have confirmed and corrected it as unpaired t-test. We have also analyzed the data using Mann-Whitney U test which also showed significance similarly with T-test.
- “In contrast, IL-7R-α subunit at mRNA and protein levels were increased in colon cancer tissues compared to matching normal tissues.” At here, I don’t think “in contrast” is a correct expression. If one thing is in contrast to another, it is very different from it.
Reply: Thank you. We have removed in contrast word from the sentence.
- “The increase of TSLP and TSLP receptors expression is closely associated with an increase of inflammation” The author should put this sentence after the last sentence. This is because the statement and evidence before the last sentence cannot deduce the association between TSLP and the increase of inflammation.
Reply: We have placed the sentence as suggested by the reviewer

Round 3
Reviewer 1 Report
Thank you for the author's reply and explanation. Please double-check the and incorrect grammar and “fold change” in your article before final submission.